# Commonsense Reasoning for Question Answering with Explanations

**Wenting Zhao** and **Alexander M. Rush** and **Claire Cardie**
Department of Computer Science
Cornell University
{wzhao,cardie}@cs.cornell.edu, arush@cornell.edu

## Abstract

Commonsense reasoning is an important capability for a range of AI applications such as text understanding. Neural models for commonsense reasoning QA often directly predict answers based on learned representations of language. In this work, we consider the challenge of producing an explicit reasoning step for a commonsense QA system. We propose a latent-variable model that identifies what type of knowledge from an external knowledge base may be relevant to answering the question, computes the commonsense inferences, and predicts the answer. Our method can therefore learn to provide posterior rationales for why a certain answer was chosen. Experimental results show that the model can identify the correct reasoning step in twice as many examples compared to an existing unsupervised approach for producing explanations, while still maintaining comparable accuracy to end-to-end pretrained models.

## 1 Introduction

Commonsense is knowledge that is considered obvious to most humans. Commonsense reasoning uses this knowledge to solve complex reasoning tasks (Sap et al., 2020; Cambria et al., 2010). Specifically, we study multiple-choice QA (MCQ) that requires commonsense reasoning. Recent approaches have applied end-to-end pretrained language models (PLMs) to solve MCQ. A downside of the approaches is that it is impossible to extract the explicit reasoning steps used by the model. To get around this issue, Bansal et al. (2021); Paranjape et al. (2021) proposed to directly predict intermediate steps in the reasoning chain. However, these methods require direct supervision on the reasoning steps, which implies manual annotations. Bosselut et al. (2021) developed an unsupervised approach to obtain explanations by leveraging a dynamic knowledge base (KB). However, because

this approach does not involved any learning component with respect to the target task, its ability to identify reasoning steps is limited.

In this work, we consider the problem of learning the reasoning path for MCQ that requires commonsense reasoning, without sacrificing the benefits of pretrained neural models. Explicitly, we propose a structured latent-variable approach that can learn the intermediate reasoning step for answering a question without supervision. Our model first identifies what type of knowledge from an external KB may be relevant, then obtains that knowledge from the KB; finally, the model predicts an answer.

We empirically evaluate our method on the socialIQA dataset (Sap et al., 2019) and show that we are able to achieve similar accuracy to that of a pretrained model while we identify the explanations. We also introduce a new evaluation set that annotates the correct reasoning steps drawn from comet2020 (Hwang et al., 2021) for test examples in socialIQA (Sap et al., 2019). On this new evaluation set, we analyze the generated explanations and show our model is able to find the correct reasoning steps in 45% cases compared to 22% for the dynamic KB method.

## 2 Related Work

**Learning explanations for commonsense reasoning.** Several multi-stage models have been proposed to produce explanations for commonsense MCQ problems. Bansal et al. (2021) first trained a model to infer free-form commonsense from the context; then they used a separate model to predict the answer conditioned on both the context and the commonsense. Paranjape et al. (2021) learned to generate contrastive commonsense explanations for coreference resolution. We note that both methods are supervised and require manually provided explanations. Additionally, Shwartz et al. (2020) hand-crafted a number of commonsense knowledge templates, and the templates were later filled

by pretrained models, which could be viewed as an explanation for choosing an answer.

Finally, Lin et al. (2017) developed a similar generative approach for a machine reading comprehension problem. They mined heterogeneous knowledge from different sources and identified the reasoning trajectory by using an attention mechanism over the mined knowledge. However, we assume different data generating processes and employ different mechanisms to incorporate KBs.

**Incorporating external knowledge sources.** Because leveraging external knowledge sources is a key component in generating explanations in our approach, we also provide an overview of different ways to incorporate such sources into QA systems. Bauer et al. (2018); Lin et al. (2019); Feng et al. (2020); Paul and Frank (2019); Yasunaga et al. (2021); Wang et al. (2020) extracted entities in contexts/questions/answers and built knowledge graphs on the entities according to external KBs; some of them made specific network architecture changes to include this information. Bauer and Bansal (2021) developed a method to choose the KB that has knowledge most aligned with the target task given several candidate KBs. Xia et al. (2019) performed multi-task learning, where, in addition to the original task, they added hand-crafted auxiliary tasks based on nodes and edges in the KB to improve generalization. The aforementioned approaches all use static KBs, which only have fixed nodes. Bosselut et al. (2021) instead utilized a dynamic KB to retrieve knowledge relevant to the context. Due to its strong generalization capacity, we use the method and adapt it to generate commonsense inferences.

## 3 Method

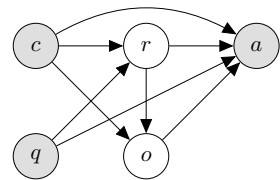

Figure 1: The graphical model for the generative process for performing commonsense reasoning. Here, $(c, r, o)$ is a knowledge triplet, where the subject is set to be the context $c$, $r$ is the relation, and $o$ is the object.

The goal of this work is to generate explanations for MCQ that requires commonsense reasoning.

We consider how humans generate explanations from the psychological point of view: given an event that a person believes has happened, the person come up with a set of explanations and select the one that best explains the event (Gordon and Hobbs, 2017). For example, we know that "Jack needed some money, so he went and shook his piggy bank. He was disappointed when it made no sound." When we are asked to explain the reason for the pippy bank made no sound, and the two possible commonsense explanations are "the piggy bank's physical material makes no sound" and "there is not money in the pippy bank." Because the latter better explains the event in that making no sound disappointed Jack, we believe this explanation is correct. We follow this procedure to produce explanations for humans to perform evaluation: we first let humans believe the MCQ example (with the answer produced by the model) is a past event, and ask humans to choose an explanation that best explains the event. Therefore, we consider the posterior rationales to be the explanation. The existing literature also refers to the process of generating commonsense explanations for given events as abductive commonsense reasoning (Bhagavatula et al., 2019).

Our approach for computing the posterior explanations will be to utilize a generative model that first retrieves knowledge relevant for a given context from an external KB. In particular, we will use Resource Description Framework (RDF) triples (Auer et al., 2007; Bollacker et al., 2008) to represent commonsense knowledge. For a given MCQ example, we can then utilize this generative model to infer the explicit reasoning used by the system on specific commonsense examples.

At a high level, we assume there is unobserved commonsense knowledge that is necessary for reaching the correct answer. However, there is a large number of commonsense tuples that may be relevant, so we need to identify the specific one that is required given the context and the question [1]. Therefore, the goal of our model is to find this missing reasoning step and return it as the explanation for the correct answer. We walk readers through the generative process with examples shown in Table 1. End-to-end neural models directly produce an answer distribution based on contexts and questions; however, we generate a reasoning step from con-

---

[1]There may be a sequence of commonsense tuples that are required for answering a question. We do not address such cases here and leave them to the future work.

| Context & Question | Reasoning step (r, o) | Answers |
|---|---|---|
| Carson brought the spoon to Taylor's mouth so Taylor could eat.

What does Carson need to do before this? | Desires to be helpful
HasSubevent PersonY is eating
MotivatedByGoal they are full
xEffect gets thanked
➡ **xNeed to be near someone** | a) bring a cup
b) leave the house
**c) sit with Taylor** |
| Lee found the Northeast to be way too cold. Lee decided to move to Florida.

How would you describe Lee? | ➡ **Desires to get away from the cold**
HasSubevent get a new job
MotivatedByGoal cold
xEffect gets cold
xNeed to have a job | a) happy
b) likes cold weather
**c) likes the heat** |
| Riley wanted to get a raise and started to work very hard at work.

What will Others want to do next? | Desires to be successful
HasSubevent get a raise
MotivatedByGoal they get a raise
➡ xEffect gets promoted
xNeed to be a good employee | **a) work hard as well**
b) quit work
c) be alone |
| Tracy sat down next to Ash and began softly kissing him.

What will Ash want to do next? | ➡ Desires to show affection
HasSubevent get closer
MotivatedByGoal none
**xEffect gets kissed back**
xNeed to be near him | a) be romantic
**b) kiss back**
c) run away |

Table 1: An MCQ example from the socialIQA dataset and possible reasoning steps extracted from an external knowledge graph. ➡ points to the reasoning step that our latent-variable model chooses to predict the answer. A reasoning step consists of a relation $r$ and an object $o$. The bold texts are the correct reasoning step and the correct answer annotated by humans. On the first two rows, the LVM is able to identify the correct knowledge; on the third row, none of the commonsense inferences retrieved from COMET can lead the correct answer; and on the last row, the LVM chooses the wrong reasoning step when a correct one is present.

texts and produce the the answer distribution also based on this intermediate step. In the first example of Table 1, the reasoning step the generative model produces is that Carson needs to be near someone in order to make the event in the context happen, which helps to reach the answer. More specifically, the generative model produces this reasoning step in two stages: the first stage identifies the relation (in blue) that may be relevant for the context and the question, and the second stage generates the most plausible object (in brown). Note that compared to prior work like (Bosselut et al., 2021), the prior work only produces commonsense inferences as intermediate step, but our model also selects the best explanation among all other explanations based on the learned representation of the context and the question.

Formally, MCQ problems start with a context $c$ and question $q$. The goal is to produce a distribution over answer strings $a$, defined by $P(a|c, q)$. To model this distribution we will introduce a latent explanation in the form of a partial RDF triple $z = (r, o)$, such that $\sum_z P(a, z|c, q)$. The complete RDF triplet has form $(s, r, o)$ where $s, o \in \mathcal{V}^*$ are a subject and an object, respectively, and $r \in \mathcal{R}$ is the relation between $s$ and $o$. $\mathcal{V}$ is a vocabulary, and $\mathcal{R}$ is a fixed set. In the MCQ task, we set $s$ to

be $c$, as $c$ can be viewed as an event, and one can make commonsense inference from the event with different relations.

Figure 1 shows the generative process, which proceeds in three stages:

$$
\begin{array}{lll}
r \sim & P(r \mid c, q) & \text{Relation Model} \\
o \sim & P(o \mid r, c) & \text{Object Model} \\
a \sim & P(a \mid c, r, o, q) & \text{Answer Model}
\end{array}
$$

The following three sections describe each of these stages in more detail.

### 3.1 Relation Model

When generating a reasoning step, the relation model determines what type of commonsense from the external KB is required to answering the question under the context. For example, when one has a question asking "how would you describe X," the commonsense that usually come up in one's mind is X's physical attributes if X is an object or X's characteristics if X is a person. Including the context further reduces ambiguity because without a scope, a question could have many interpretations. Therefore, the relation model specifies a distribution over all relations conditioned on $c$ and $q$. We

parameterize $P(r|c, q; \theta)$ by a BERT model which takes in $[CLS]\ c\ [SEP]\ q\ [SEP]\ r$. $r$ is the relation symbol from the relation set defined by the external knowledge base (and this also applies to the later models where $r$ is also part of the input).

## 3.2 Object Model

After the type of relevant knowledge is identified, the object model then generates commonsense inferences given the context and the relation. Learning to infer the commonsense knowledge from a context and a knowledge type is in fact a well-studied problem, called KB completion (Saito et al., 2018; Malaviya et al., 2020). We thus treat it as a KB completion task. The object model specifies a distribution over all objects $o$ conditioned on $c$ and $r$. $P(o|c, r; \phi)$ is parameterized by a BART model (Lewis et al., 2020), where the input is a concatenation of $c$ and $r$.

## 3.3 Answer Model

We arrive at the final component of our generative model, which governs how the information about contexts, questions, and knowledge are rendered into answers. The answer model explicitly considers the commonsense inferred from the context by conditioning on the RDF triplet and produces a probability distribution over all answers. $P(a|c, r, o, q; \psi)$ is also parameterized by a BERT model with a multiple-choice head. The input to the model is $[CLS]\ c\ [SEP]\ r\ [SEP]\ o\ [SEP]\ q\ [SEP]\ a$.

## 3.4 Training and Inference

The generative model is trained in two steps. It first learns the object model; then, it uses the following objective to jointly learn the relation model and the answer model, summing out $r, o$:

$$\max_{\theta, \psi} \sum_{r, o} P(a|c, r, o, q; \psi) P(o|c, r) P(r|c, q; \theta).$$

Because $\mathcal{V}^*$ is a combinatorially large space, exactly enumerating all objects is intractable. The joint distribution is then approximated by:

$$\sum_{r} \max_{o} P(a|c, r, o, q; \psi) P(o|c, r) P(r|c, q; \theta),$$

Note that we sum over all $r$'s, and for each $r$ we apply greedy search to pick the $o$ that maximizes the likelihood. To get an explanation from the

| Type | Form | Relation |
|------|------|----------|
| wants | What will X want to do next? | xWant
oWant
HasSubEvent |
| reactions | How would X feel afterwards? | xReact
oReact
Cause |
| descriptions | How would you describe X? | xAttr |
| motivations | Why did X do this? | xReason
HinderedBy
xIntent |
| needs | What does X need to do before this? | xNeed
isFilledBy
isAfter |
| effects | What will happen to X? | xEffect
oEffect
isBefore |

Table 2: A rule-based baseline for commonsense reasoning. Since the questions in socialIQA are categorized in six types, a simple rule-based baseline can choose a fixed set of relations based on the question form.

model, we compute a posterior rational as follows:

$$P(r|c, q, a) = P(r|c, q) \cdot \frac{\sum_o P(a, o|c, q, r)}{\sum_{r', o} P(a, r', o|c, q)}$$

## 4 Experiments

The goal of the system is to identify the intermediate reasoning step used in question answering. We therefore experimentally evaluate two aspects of our model – accuracy of answers and correctness of reasoning steps.

## 4.1 Setup

**Datasets.** The relation model and the answer model are trained on the socialIQA dataset (Sap et al., 2019). SocialIQA has 37,588 multiple-choice questions that covers the pragmatic implications of everyday, social events. The object model is trained on ATOMIC2020 (Hwang et al., 2021), which consists of 1.33M RDF tuples about common entities and events with 23 unique relation types. However, the relation model only places a distribution over the 16 relations related to social interactions.

**Baselines.** The baseline for comparing accuracy is a standard BERT base model with a multiple-choice head (Sap et al., 2019) (referred to as BERT). Two baselines are considered for evaluating the reasoning steps. Table 2 shows the first baseline where each question type associates with a set of relations (referred to as rule-based). The second baseline uses the object likelihoods provided by

|  | valid | test |
|---|---|---|
| BERT (original) | 63.30% | 63.10% |
| BERT (reproduced) | 62.23% | 62.28% |
| Ours | **62.74%** | **63.35%** |

Table 3: Accuracy for each approach on *socialIQA*. Original BERT accuracy is reported in Sap et al. (2019). Reproduction and our generative model use the same framework.

|  | Relation ACC | RDF ACC |
|---|---|---|
| Rule-based | 31.4% | 13.6% |
| COMET | 22.2% | 22.2% |
| Ours | **55.6%** | **45.6%** |

Table 4: Evaluation for identifying the reasoning steps. Relation accuracy reflects if an approach has chosen the correct knowledge types, and RDF accuracy reflects if an approach has found the full reasoning steps.

COMET (Bosselut et al., 2021) for choosing the most likely commonsense inference (referred to as COMET).

**Implementation & Hyperparameters.** We implement both latent variable and non-latent variable (baseline) models with BertModel in Huggingface Transformers (Wolf et al., 2020). For the object model, we use the released BART-large model on github. [2] We perform grid search with learning rates {5e-6, 1e-5, 3e-5, 5e-5} and batch sizes {1,2,3,4,8} to achieve best-possible performance for the baseline model. Due to limited computation resources, we only fine-tune the latent variable model with a learning rate of 1e-5 and batch sizes of {1,2,3}. For both models, we warm up the learning rate for first 10% steps and train for five epochs.

**Evaluation metrics.** To check the correctness of the reasoning steps, we annotate 500 test examples in socialIQA: for each example, we label up to three relations if they may lead to useful objects that help to reach the answer. Furthermore, we also annotate if their subsequent objects are correct; that is, the entire RDF triple is correct. Each approach is allowed to choose three relations. If any of the relations is correct, then it is considered to identify the correct knowledge type; if the objects followed from the relations are also correct, then it is considered to find the fully correct reasoning step.

### 4.2 Results

Table 3 summarizes the accuracy results. For each approach, the test result obtained from evaluating the checkpoint with the best validation accuracy is reported. Our model achieves an accuracy of 63.13% whereas the baseline is 62.28%. Therefore, the latent-variable model is able to maintain similar accuracy to the pretrained model.

Table 4 shows the results on identifying reasoning steps. For choosing a relation, our model improves 24.2% over the rule-based method, suggest-

---

[2]github.com/allenai/comet-atomic-2020

ing that our generative model has learned how to identify the relevant knowledge type for commonsense examples. Furthermore, when comparing the RDF accuracy (i.e., the reasoning step is fully correct), our model improves 23.4% over COMET, serving as an evidence that our relation model and answer model have learned explanations specific to the task. In summary, our model is able to identify the correct reasoning step that is not explicitly present in the context but is required to derive the answer for a substantial number of cases without supervision.

## 5 Conclusion

We propose a latent-variable model to generate explanations for commonsense reasoning QA without supervision. The experimental results show that our approach achieve similar accuracy to the pretrained model. The human evaluation suggests our model can identify the correct reasoning steps for significantly more examples than an existing unsupervised approach for producing explanations.

## 6 Acknowledgments

We would like to thank Justin Chiu and Celine Lee for valuable feedback.

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
