# OpenReview forum: "Commonsense Reasoning for Question Answering with Explanations"
_aclweb.org/ACL/2022/Workshop/CSRR — ACL 2022 Workshop CSRR_

### Official Review · Reviewer_GP1G · 2022-03-23

**Rating:** 6
**Confidence:** 3

**Review:**

# Summary
This work tackles the problem of explainable commonsense QA. They propose a latent-variable model that identifies what type of knowledge from an external knowledge base may be relevant to answering the question, computes the commonsense inferences and predicts the answer. The method can therefore learn to provide posterior rationales for why a certain answer was chosen. Experimental results show that the model can identify the correct reasoning step in twice as many examples compared to an existing unsupervised approach for producing explanations for the socialIQA task, while still maintaining comparable accuracy to end-to-end pretrained models.

# Contributions
1. Propose an explainable latent-variable commonsense QA model.
2. Perform experiments to show that the model can accurately identify the reasoning steps for the socialIQA task, while also maintaining predictive accuracy.

# Pros
1. Tackle an interesting and important problem: explainability for commonsense reasoning.
2. Thorough evaluation of the proposed model, including manually annotating the groundtruth reasoning steps of 500 examples in socialIQA test sets and evaluating the model's accuracy in identifying those reasoning steps.
3. The proposed generative model is novel.
4. The proposed model can accurately identify ground-truth reasoning steps.

# Cons
1. The writing is not that clear and the paper is hard to follow in general. Working through an example instead of just showing this plate notation in Figure 1 can help. Restructuring and rewriting of the paper are needed.
2. I did not get the general message of the paper until I read it several times. I suggest in your writing you stress that the paper proposes a commonsense QA model that is **explainable**.  At the end of day, the paper proposes a model for commonsense QA. The ability of generate explanations is just one of its characteristics.
3. The arguments that humans first reach an answer then work backwards to figure out the commonsense knowledge they use on line 122-132 are not that convincing to me. Maybe switch to a better example.
4. Not sure what the "KB-based" baseline is for Table 3 even after reading the description on line 251-254. Does "external KB" here mean COMET?
5. No standard deviations are reported in Table 2.

# Other comments and questions
1. On line 158, what does "such that \sum_zP(a,z|c,q)" mean?
2. On line 163-166, I am not sure how this computation is done even if I have read the Bosselut et al 2021 paper.
3. The citation on line 189 is misformatted.
4. Notation inconsistency for V* (line 159, 161 and 220)

---

### Official Review · Reviewer_F2Z9 · 2022-03-23
**Interesting work but can be improved with clearer presentation**

**Rating:** 6
**Confidence:** 4

**Review:**

This work proposes a method for explainable question-answering for SocialIQA. The explanations are retrieved from ATOMIC, which contains (subject, relation, object) tuples that can provide helpful evidence/explanation for answering the question. The method trains a pipeline of generative models: i) for generating a relevant relation r in ATOMIC that could be useful for predicting the answer, ii) generating the object using r; and finally, iii) generating the answer using the generated + question and context.

The three-step approach achieves results comparable to a pre-trained model, but has the added advantage that it can provide the tuples used for answering the question, possibly adding to the explainability of the model.

The paper presents an interesting case of using KB information for commonsense question answering.
However, the present version contains several shortcomings in both formulation and experimental design, listed next.

* Trivial baselines: The baselines for retrieving knowledge from the external KB are pretty weak. For example, we can train a simple classifier for relation prediction by treating context in SocialIQA and subject in ATOMIC as exchangeable. Then, the prediction of the classifier can be used as an explanation. It appears that such an approach will work, and might be worth experimenting with in the future.

* Clarity of presentation: Several important details are unclear. For instance, ATOMIC has tuples (s, r, o), and each QA sample is (c, q, a). As the authors mention, for SOCIAL-IQA, c determines s, but r and o are unknown. However, the method description mentions that the "object model is learned first." How can the object model be learned without (r, c, o) tuples? This also holds for learning the relation model. I suspect it is trained by simply treating s == c in ATOMIC, but this is an important detail that should be made more explicit. Please also see possible corrections in the formulation.




* Novelty: the main idea of leveraging external knowledge for commonsense reasoning has been explored in several works. The primary differentiation seems to be that the proposed method does not require hand-annotated explanations and uses dynamic KB. The difference with Bosselut 2021 (which the authors mention) is not completely clear. Further, see [1] for an approach that does not require explicit annotation and does not rely on fixed KBs. Overall, better positioning concerning the related work will help understand the core contributions of this work.



### Possible corrections in the formulation:

1. In the max_ equation, the object model should also condition on the question for the chain rule factorization to work. Currently, there is an implicit assumption that o is independent of q given c, which may or may not be valid. Consequently, the authors may try conditioning on the question in the object model as well. It might improve things! As a side remark, V* is not specified. The notation also changes from $\theta$ to $\theta’’$ without any explanation.

2. L224: "To get an explanation…." The following equation appears to be incorrect. First, isn't z = (r, o), and thus the explanation should contain both the terms? Second, o is introduced in the equation on the RHS, but there is no marginalization over it. I suspect there's a typo/problem with the formulation.





### Questions:

1. L130: Due to 130 this ambiguity, humans tend to perform explicit 131 reasoning afterwards. Therefore, we consider the 132 explanation to come after an answer being chosen.

What are the implications of this statement for your method? I don't think that the answer goes back to generating the input?


Overall, I think the work highlights an important area of commonsense reasoning and thus will be a useful addition to the workshop. However, I hope the paper improves by adding the clarifications mentioned above.

---

[1] Madaan, Aman, Niket Tandon, Dheeraj Rajagopal, Peter Clark, Yiming Yang, and Eduard Hovy. "Think about it! Improving defeasible reasoning by first modeling the question scenario." In Proceedings of the 2021 Conference on Empirical Methods in Natural Language Processing, pp. 6291-6310. 2021.

---

### Official Review · Reviewer_anxb · 2022-03-25
**Interesting paper of producing reasoning steps for QA**

**Rating:** 7
**Confidence:** 4

**Review:**

Strengths
- The paper describes an interesting approach to producing an explanation, in the form of a knowledge base tuple, for a QA answer. This is potentially interesting to not only the commonsense reasoning community but also people interested in understanding the explicit reasoning steps behind model answers
- The model achieves strong performance on two evaluation settings

Cons
- It is hard to determine the quality of the manual annotation and the output without seeing some examples, and there appears to be only one in the paper.
- There are some experimental details that should be clarified, especially on the input/output of the different components

Questions:
L165: what is this way? It should be described here, so the paper is self-contained.
L181: What would the actual relation from a database be here?
3.3.: Not completely clear what the answer model is actually producing
3.2: Do you use greedy search to select a relation to use here? Or do you compute the object model over all possible relations?
L249: how do you get the explanation from the set of relations? how does the second baseline choose the most likely inference? I think I understand some about the baselines from looking at the results, but the actual description needs to be clarified when they are introduced.
L269: do you compute any kind of agreement here? how do you check for quality?

Typos:
L180-181: "the commonsense usually" --> "the commonsense that usually"
L182: "physical entities" --> "physical attributes"
L243: should this be "object model" not "relation model"?

---

### Decision · Program_Chairs · 2022-03-28

Accept